# *Pseudomonas aeruginosa* Nonphosphorylated AlgR Induces Ribonucleotide Reductase Expression under Oxidative Stress Infectious Conditions

Alba Rubio-Canalejas,[a] Joana Admella,[a] Lucas Pedraz,[a] 🄳 Eduard Torrents[a,b]

[a]Bacterial Infections and Antimicrobial Therapies Group, Institute for Bioengineering of Catalonia (IBEC), Barcelona Institute of Science and Technology (BIST), Barcelona, Spain
[b]Microbiology Section, Department of Genetics, Microbiology and Statistics, Faculty of Biology, University of Barcelona, Barcelona, Spain

**ABSTRACT**    Ribonucleotide reductases (RNRs) are key enzymes which catalyze the synthesis of deoxyribonucleotides, the monomers needed for DNA replication and repair. RNRs are classified into three classes (I, II, and III) depending on their overall structure and metal cofactors. *Pseudomonas aeruginosa* is an opportunistic pathogen which harbors all three RNR classes, increasing its metabolic versatility. During an infection, *P. aeruginosa* can form a biofilm to be protected from host immune defenses, such as the production of reactive oxygen species by macrophages. One of the essential transcription factors needed to regulate biofilm growth and other important metabolic pathways is AlgR. AlgR is part of a two-component system with FimS, a kinase that catalyzes its phosphorylation in response to external signals. Additionally, AlgR is part of the regulatory network of cell RNR regulation. In this study, we investigated the regulation of RNRs through AlgR under oxidative stress conditions. We determined that the nonphosphorylated form of AlgR is responsible for class I and II RNR induction after an $H_2O_2$ addition in planktonic culture and during flow biofilm growth. We observed similar RNR induction patterns upon comparing the *P. aeruginosa* laboratory strain PAO1 with different *P. aeruginosa* clinical isolates. Finally, we showed that during *Galleria mellonella* infection, when oxidative stress is high, AlgR is crucial for transcriptional induction of a class II RNR gene (*nrdJ*). Therefore, we show that the nonphosphorylated form of AlgR, in addition to being crucial for infection chronicity, regulates the RNR network in response to oxidative stress during infection and biofilm formation.

**IMPORTANCE**    The emergence of multidrug-resistant bacteria is a serious problem worldwide. *Pseudomonas aeruginosa* is a pathogen that causes severe infections because it can form a biofilm that protects it from immune system mechanisms such as the production of oxidative stress. Ribonucleotide reductases are essential enzymes which synthesize deoxyribonucleotides used in the replication of DNA. RNRs are classified into three classes (I, II, and III), and *P. aeruginosa* harbors all three of these classes, increasing its metabolic versatility. Transcription factors, such as AlgR, regulate the expression of RNRs. AlgR is involved in the RNR regulation network and regulates biofilm growth and other metabolic pathways. We determined that AlgR induces class I and II RNRs after an $H_2O_2$ addition in planktonic culture and biofilm growth. Additionally, we showed that a class II RNR is essential during *Galleria mellonella* infection and that AlgR regulates its induction. Class II RNRs could be considered excellent antibacterial targets to be explored to combat *P. aeruginosa* infections.

**KEYWORDS**    ribonucleotide reductase, biofilm, oxidative stress, AlgR, *Galleria mellonella*, *nrdJ*

Address correspondence to Eduard Torrents, etorrents@ibecbarcelona.eu.

The authors declare no conflict of interest.

**P**seudomonas aeruginosa is a versatile Gram-negative opportunistic pathogen which causes acute and chronic illness in immunocompromised patients and people with cystic fibrosis (CF), chronic obstructive pulmonary disease, and other diseases. In

addition, it is known as one of the major sources of pulmonary health care-associated infections and the primary cause of morbidity and mortality (1). Once infection begins, *P. aeruginosa* colonizes the lungs in its nonmucoid form. However, as the infection course continues, *P. aeruginosa* switches into a mucoid phenotype involving massive alginate production, increasing lung deterioration and leading to a poor prognosis (2).

One of the leading players involved in the production of alginate and thus in the chronicity of infection is the *algD* operon, which encodes the main enzymes responsible for alginate production. The *algC* gene, located within the *algC-argB* operon, encodes an enzyme required for alginate biosynthesis and LPS production (3). Alginate production is a very advantageous form of bacterial protection because it prevents phagocytosis, antibiotic penetration, and desiccation. The metabolic process of alginate production is highly regulated because it is highly energy consuming (4).

One of the most significant regulators of alginate production is the *algU/mucABCD* operon. AlgU is an alternative sigma factor ($\sigma^E$) which, in the nonmucoid phenotype, is sequestered by the anti-sigma factor MucA. During an infection, the cellular stress produced by the immune system induces the proteolytic degradation of MucA, releasing AlgU and therefore activating the transcription of several genes, such as the *algD* operon, promoting alginate synthesis (5) and the expression of *algR* (6).

AlgR is a transcription factor that is part of the two-component system FimS-AlgR with FimS, a membrane kinase which, in response to external signals, can phosphorylate AlgR. AlgR is a transcription factor which functions as a global regulator, controlling several metabolic pathways depending on its phosphorylation state. It is known that phosphorylated AlgR activates rhamnolipid formation, cyanide production, and type IV pili, which are important for twitching motility. It seems that the biological pathways activated by phosphorylated AlgR are related to the first phases of infection, regulating functions related to cell attachment and initial biofilm formation (3). On the other hand, nonphosphorylated AlgR activates important pathways of biofilm formation, such as the production pathway of the polymer alginate (7). Thus, the metabolic pathways triggered by AlgR in its nonphosphorylated state are related to late biofilm formation, including alginate biosynthesis and the mucoid phenotype which lead to chronic infections (3).

In addition, we and other authors have linked the *algR* system with the *P. aeruginosa* ribonucleotide reductase (RNR) network (8). RNRs are key enzymes which catalyze the reduction of ribonucleotides (NTPs) to deoxynucleotides (dNTPs), which are essential for DNA repair and *de novo* synthesis (9). RNRs are classified into three different classes (I, II, and III) based on their three-dimensional structure, use of cofactors and metals, dependence on oxygen, and radical generation. Class I RNRs are encoded by *nrdAB*, and the enzymes produced are only active under aerobic conditions. Class I is subsequently classified into subclasses Ia, Ib, Ic, Id, and Ie based on the metal center requirement to form the protein radical (10). Class II RNRs, encoded by *nrdJ*, require vitamin $B_{12}$ to be enzymatically active, and their activity is oxygen-independent (11). Class III RNRs, encoded by *nrdDG*, require anaerobic conditions to be active (12). *P. aeruginosa* encodes all three RNR classes, which is very advantageous for its survival in different environmental conditions (9, 13). Depending on the growth and infection phase, specific RNRs are activated; for example, we previously showed that *nrdJ*, encoding class II RNRs, is highly expressed under anaerobic conditions and during biofilm growth (11, 12, 14).

The macrophages and neutrophils of the host immune system try to remove the pathogen by producing reactive oxygen species (ROS) which cause oxidative stress (15). ROS can cause damage to DNA, lipids, proteins, and nucleic acids; thus, bacteria have several ways to protect themselves from oxidative stress (15). *P. aeruginosa* has two superoxide dismutases, three catalases (KatA, KatB, and KatC), and four alkyl hydroperoxide reductases to eliminate the ROS that produce oxidative stress (15). A biofilm formed with alginate protects the bacteria from antibiotics and the host immune system (4). Additionally, bacteria embedded within a biofilm acquire protection from oxidative stress (16).

Previous studies have revealed the relationship between FimS-AlgR and the RNR. We previously described that the AlgR regulation of class I and II RNRs depends on its phosphorylation state (8). Additionally, AlgR is linked with the oxidative stress response system (17), and it is known that the expression of the *Escherichia coli* RNR is activated under oxidative stress conditions (18). We have also discovered that AlgR is the transcription factor responsible for the induction of *P. aeruginosa* class I and II RNRs under oxidative stress (8).

In this study, we delved into the regulation of class I and II RNRs by AlgR in the presence of $H_2O_2$ in different biological growth stages, in a planktonic culture, in a biofilm, and during *G. mellonella* infection. To determine gene regulation variability, we measured RNR expression in other *P. aeruginosa* strains (laboratory and clinical isolates). To obtain a clinical perspective on RNR expression under oxidative conditions, one of the strains used was a clinical strain isolated from a CF patient (12). In this work, we confirmed that AlgR, in its nonphosphorylated state, is responsible for class I and II RNR transcriptional induction under oxidative stress conditions. We demonstrated that AlgR is essential for *nrdA* and *nrdJ* expression in biofilms under stress conditions. Finally, we observed that ROS are produced during *G. mellonella* infection and that *nrdJ* is the principal RNR expressed, with AlgR being essential for its induction. The different *nrdJ* and *nrdA* induction patterns help to elucidate the essential role of NrdJ in ensuring bacterial survival and suggest a molecular pathway used by *P. aeruginosa* during bacterial infection to restore the dNTP pool and repair damaged DNA.

## RESULTS

**AlgR expression is induced under oxidative stress conditions.** AlgR is a transcription factor that is part of the *fimS-algR* operon. FimS-AlgR is a two-component system of *P. aeruginosa* that can regulate several metabolic pathways in response to external signals. However, the expression of *algR* can be activated through two different promoter regions, P*fimS* and P*algR* (7). Figure 1A shows a scheme of the two promoters in *fimS-algR*. The first promoter is located upstream of *fimS*; when it is transcriptionally activated, the transcript produced encodes the FimS and AlgR proteins. The second promoter is located upstream of *algR* within the *fimS* coding region. This second promoter responds to specific sigma factors, such as AlgU and RpoS, to activate its transcription, and the mRNA obtained encodes only the protein AlgR (7).

Previous studies have linked AlgR with the bacterial response system to oxidative stress (8, 19). However, the specific mechanism by which AlgR is activated under oxidative stress conditions has not been clarified. In this work, we aimed to examine the direct link between AlgR and oxidative stress. First, we transcriptionally fused the P*fimS* promoter and the P*algR* promoter to *gfp*, obtaining pETS-P*fimS* and pETS-P*algR*, respectively (see Materials and Methods). Our goal was to determine whether the expression of *algR* was induced under oxidative stress conditions and, if so, which promoter was responsible for this specific activation. Figure 1B shows that during the exponential phase ($OD_{600}$ [optical density at 600 nm] = 0.5), *algR* expression slightly increased after $H_2O_2$ was added. When *algR* expression was measured in the stationary growth phase ($OD_{600} > 2.5$), P*algR* had a higher basal expression level (2.7 times higher) than P*fimS*. In addition, only pETS-P*algR* displayed significantly increased *algR* expression after the addition of $H_2O_2$. To test our system and as a control, we transcriptionally fused the promoter regions of two *P. aeruginosa* catalases, *katA* and *katB*, to *gfp* and analyzed whether the $H_2O_2$ concentration used was suitable for generating oxidative stress. The catalases *katA* and *katB* are part of the oxidative stress response system in *P. aeruginosa*. *katA*, the main catalase of *Pseudomonas*, is constitutively expressed to remove the $H_2O_2$ produced in the bacterial respiratory network. When ROS levels are high, other proteins, such as KatB, are expressed to remove excess oxidative stress (15). Figure 1C shows that both catalases, pETS-P*katA* and pETS-P*katB*, were transcriptionally induced when 1 mM $H_2O_2$ was added for 30 min during the exponential and stationary growth phases. This clearly indicated that the conditions used in this study created enough ROS to generate oxidative stress growth conditions.

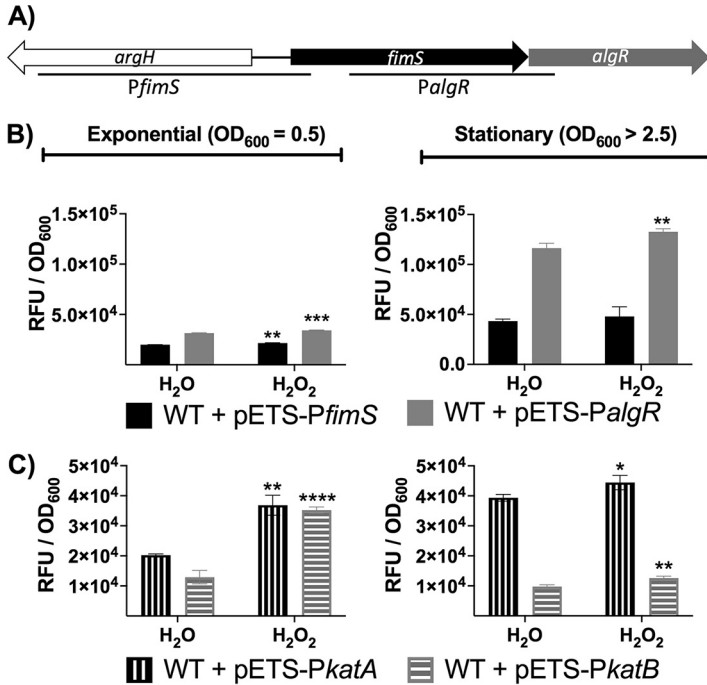

**FIG 1** Induction of *algR* and catalases in *Pseudomonas aeruginosa* under oxidative stress. (A) Scheme of the *algR* genetic background. The *algR*-specific promoters are indicated with lines. (B) Gene expression of the two different promoter regions of *algR*, P*fimS-algR* and P*algR*, which were transcriptionally fused to *gfp* (pETS-P*fimS* and pETS-P*algR*, respectively), in the exponential (OD$_{600}$ [optical density at 600 nm] = 0.5) and stationary phases (OD$_{600}$ > 2.5) after the addition of 1 mM H$_2$O$_2$ or the equivalent volume of water for 30 min. (C) The promoter regions of *katA* and *katB* were transcriptionally fused to *gfp* (pETS-P*katA* and pETS-P*katB*, respectively). The strains were incubated with 1 mM H$_2$O$_2$ or an equivalent volume of water for 30 min in the exponential (OD$_{600}$ = 0.5) and stationary phases (OD$_{600}$ > 2.5). Three independent experiments were performed; error bars indicate standard deviation. Statistical analysis to determine significant differences between the H$_2$O$_2$ and their H$_2$O counterpart samples was performed using Student's unpaired *t* test (**, $P < 0.01$; ***, $P < 0.001$).

**Nonphosphorylated AlgR controls the expression of class II (*nrdJ*) and Ia (*nrdA*) RNRs under oxidative stress conditions.** Crespo et al. (8) reported that during the exponential growth phase, RNR expression increases under oxidative stress conditions because the master transcriptional regulator AlgR directly binds to an AlgR box in the promoter region of each RNR. We aimed to elucidate how oxidative stress regulates the different RNR genes more deeply via AlgR. Additionally, because regulation by AlgR is linked to the AlgR phosphorylation state, we studied how the phosphorylated form of AlgR modulates *nrd* expression (8).

The promoter regions of *nrdA* and *nrdJ* were transcriptionally fused to the reporter gene *gfp* to obtain pETS-P*A* and pETS-P*J*, respectively. In the promoter region of *nrdA*, one AlgR box was found upstream from the coding region. In the *nrdJ* promoter region, two AlgR boxes were found (8). The RNR promoter regions with their respective mutated AlgR boxes were transcriptionally fused to *gfp* as well, yielding pETS-P*A*-Δbox1 and pETS-P*J*-Δbox1+2. These plasmids were transformed into different *P. aeruginosa* strains: PAO1 and PAO1 Δ*algR*, PA14 and PA14 Δ*algR*, and the clinical *P. aeruginosa* CF isolate PAET1 and PAET1 Δ*algR* (Fig. 2 and 3). The Δ*algR* strains were complemented with pUCP-AlgR, which encodes the wild-type (WT) AlgR protein, or with pUCP-D54N, which encodes the AlgR protein with a mutation in amino acid 54 (D54N) to prevent phosphorylation (3).

To study the expression of RNRs under oxidative stress conditions, we treated the bacterial cultures with 1 mM H$_2$O$_2$ for 30 min when the cultures reached the exponential (OD$_{600}$ = 0.5) and stationary (OD$_{600}$ > 2.5) growth phases. We observed that *nrdJ* (class II RNR) was transcriptionally induced in both the exponential and stationary growth phases (Fig. 2A). Moreover, *nrdJ* induction was eliminated in a *P. aeruginosa* PAO1 Δ*algR* strain

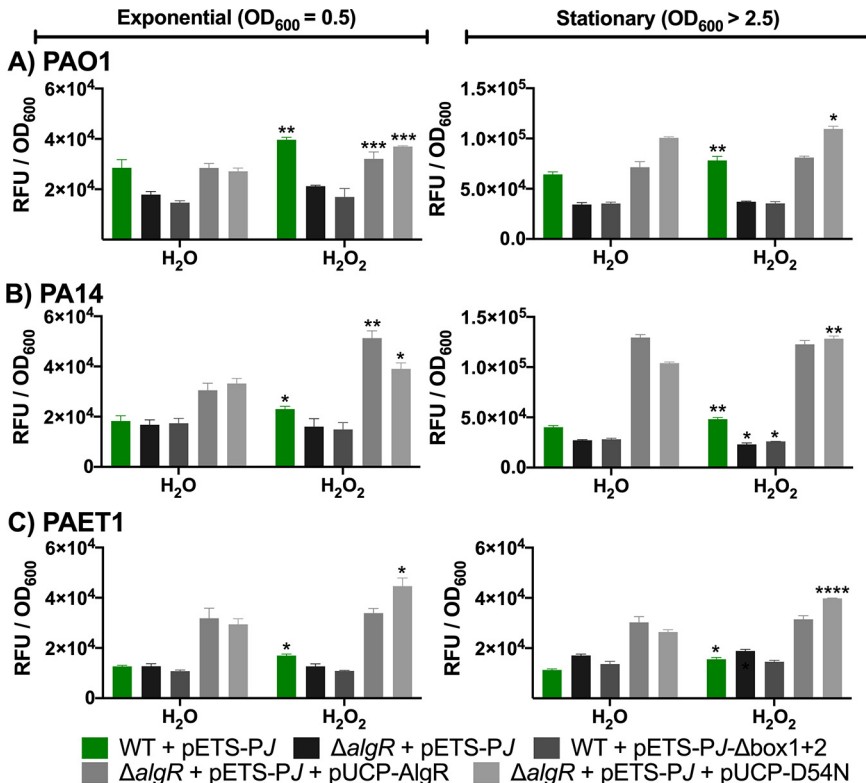

**FIG 2** Oxidative stress conditions induce *nrdJ* expression through nonphosphorylated AlgR in different *P. aeruginosa* strains. The promoter region of *nrdJ* (P*J*) was transcriptionally fused to *gfp* (pETS-P*J*). The laboratory strains PAO1 (A) and PA14 (B) and the clinical isolate PAET1 (C) were used to monitor gene expression at different time points ($OD_{600} = 0.5$ and $OD_{600} > 2.5$). The strains were incubated with 1 mM $H_2O_2$ or an equivalent volume of water for 30 min. Three independent experiments were performed, and standard deviation is indicated with error bars. Statistical analysis to determine significant differences between the samples treated with $H_2O_2$ and their counterpart samples treated with $H_2O$ was performed using Student's unpaired *t* test (*, $P < 0.05$; **, $P < 0.01$; ***, $P < 0.001$; ****, $P < 0.0001$).

and when the plasmid pETS-P*J*-Δ*box1+2* was used (Fig. 2A), indicating the specific dependence on AlgR during transcription. In addition, the complementation of the *algR* deletion with pETS-D54N restored *nrdJ* induction under oxidative stress conditions in the exponential growth phase. Both the pETS-AlgR and pETS-D54N plasmids complemented the *algR* deletion in the stationary phase. However, the induction was significant only when AlgR was not phosphorylated.

To unravel the molecular mechanism involved in the transcriptional gene induction of class I and II RNRs in strains other than the laboratory strain PAO1, we measured the expression of *nrdJ* and *nrdA* in the laboratory strain *P. aeruginosa* PA14 and the clinical strain *P. aeruginosa* PAET1. We observed that the promoter region of *nrdJ* was conserved with 99% identity, while that of *nrdA* was conserved with 100% identity, among the three strains (Fig. S1). Thus, we surmised that measurement of *nrdA* and *nrdJ* expression would provide information on how the genetic background of each strain affects the expression of class I and II RNRs under oxidative stress conditions.

When using the laboratory strain *P. aeruginosa* PA14, we observed that the basal expression level of *nrdJ* was similar to that of *nrdJ* in PAO1 (Fig. 2B). Figure 2B shows that the oxidative stress generated after incubation with 1 mM $H_2O_2$ for 30 min induced *nrdJ* expression during the exponential and stationary growth phases. AlgR caused this gene induction, as indicated by the finding that the increased *nrdJ* expression was eliminated in a PA14 Δ*algR* strain and when using the plasmid pETS-P*J*-Δ*box1+2*. Expression was restored when the gene deletion was complemented with AlgR. In the exponential phase, the induced expression of *nrdJ* was higher when pETS-AlgR was used than when pETS-

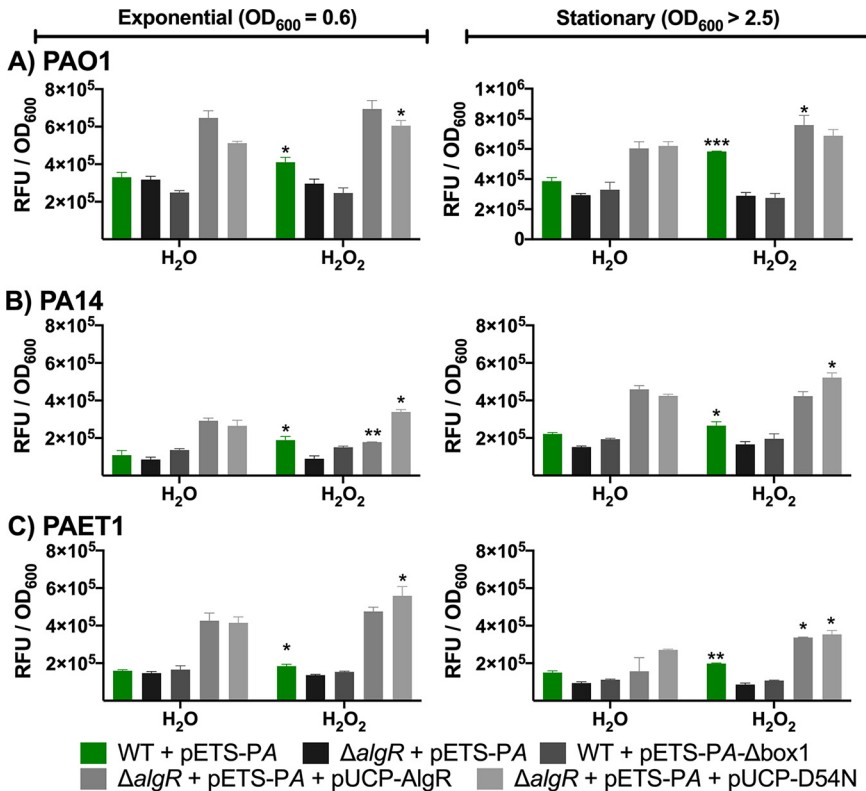

**FIG 3** *nrdA* expression is AlgR-dependent under oxidative stress conditions in several *P. aeruginosa* strains. The promoter region of *nrdA* (P*A*) was transcriptionally fused to *gfp* (pETS-P*A*). Gene expression was monitored in the laboratory strains PAO1 (A) and PA14 (B) and the clinical isolate PAET1 (C) at different time points (OD$_{600}$ = 0.6 and OD$_{600}$ > 2.5). The strains were incubated with 1 mM H$_2$O$_2$ or an equivalent volume of water for 30 min. Three independent experiments were performed, and the standard deviation is indicated with error bars. Statistical analysis to determine significant differences between the counterpart samples treated with H$_2$O$_2$ or H$_2$O was performed using Student's unpaired $t$ test (*, $P < 0.05$; **, $P < 0.01$; ***, $P < 0.001$).

D54N was used. In the stationary phase, the induction of *nrdJ* was reestablished after AlgR-D54N was used.

We also measured RNR expression in the clinical strain *P. aeruginosa* PAET1, isolated from a chronic CF patient. Figure 2C shows the basal *nrdJ* expression in the clinical isolate PAET1 and *nrdJ* induction under incubation with 1 mM H$_2$O$_2$ for 30 min. The graphs show that *P. aeruginosa* PAET1 *nrdJ* expression was lower than that in the laboratory strain PAO1. *nrdJ* expression increased after the addition of H$_2$O$_2$ in the exponential and stationary phases. The induction was abolished in the Δ*algR* strain or when the AlgR boxes of the promoter region were mutated. When the *algR* deletion was complemented with AlgR and AlgR-D54N, *nrdJ* induction was restored. When AlgR-D54N was used, the expression of *nrdJ* was higher than that when AlgR was used. This may imply that *nrdJ* induction under oxidative stress is due to nonphosphorylated AlgR.

The same type of regulation pattern was found when the general transcription of *nrdA* (class I RNR) under oxidative stress conditions was analyzed. The strains were incubated for 30 min with 1 mM H$_2$O$_2$ when they reached exponential and stationary growth phases. Figure 3A shows that *nrdA* expression increased during the exponential and stationary phases in *P. aeruginosa* PAO1. This induction was abolished in PAO1 Δ*algR*, and when we used the plasmid pETS-P*A*-ΔAlgR*box1*, which carries a mutation in its AlgR binding box, the transcriptional induction was found to be specific to the transcription factor AlgR. The *algR* deletion was complemented with AlgR-D54N, restoring *nrdA* expression.

*nrdA* showed lower basal expression in the *P. aeruginosa* laboratory strain PA14 than in PAO1 (Fig. 3B), demonstrating its strain variability. *nrdA* expression was also induced after 1 mM $H_2O_2$ was added (by 1.7 times in the exponential growth phase and 1.2 times in the stationary phase). This induction was abolished in the Δ*algR* strain and when using the plasmid pETS-P*A*-ΔAlgR*box1*, again showing that AlgR was responsible for *nrdA* induction. The induction was restored when Δ*algR* was complemented with the nonphosphorylated AlgR.

Finally, the expression of *nrdA* in the clinical isolate PAET1 is shown in Fig. 3C. The overall expression of *nrdA* was lower in PAET1 than in the PAO1 laboratory strain (by 26 times in the exponential phase and 2.5 times in the stationary phase). Even so, the induction pattern observed in the strains PAO1 and PA14 was also observed in PAET1. When 1 mM $H_2O_2$ was added, *nrdA* was induced in the exponential and stationary phases, and this induction was abolished in both the Δ*algR* strain and when the AlgR binding box was removed in the plasmid pETS-P*A*-ΔAlgR*box1*. *nrdA* induction was reestablished when the Δ*algR* strain was complemented with AlgR-D54N in the exponential phase and with AlgR and AlgR-D54N in the stationary phase.

**During biofilm growth, class II and I RNR expression under oxidative stress conditions is AlgR-dependent.** The different environmental characteristics inside a biofilm also affect the transcriptional activation of RNRs. While *nrdA* expression is activated in the upper layers of the biofilm, where oxygen is available, the class II (*nrdJ*) and III (*nrdD*) RNRs are activated in the deeper biofilm layers. Additionally, we previously demonstrated the essential role of *nrdJ* during biofilm development in *P. aeruginosa* PAO1 (8, 11, 14).

We used flow-cell chambers to grow different *P. aeruginosa* strain biofilms and measured RNR gene expression under oxidative stress conditions (see Materials and Methods). Luria-Bertani (LB) medium supplemented with 4 mM $H_2O_2$ was added to a mature biofilm (96 h) to generate oxidative stress. Negative controls were treated with LB medium. Following an incubation period of 4 h, the biofilm was dyed and imaged under a confocal microscope (see Materials and Methods). The biofilm biomass was stained with SYTO60, which appears gray in the images (Fig. 4 and 5); the DNA damage due to the oxidative stress present in the biofilm was dyed with CELLRox Orange, shown in magenta; and the RNR-specific expression was measured using green fluorescent protein (GFP) fluorescence (*nrdJ* and *nrdA*), shown in green (Fig. 4 and 5).

Figure 4A shows that *nrdJ* expression (pETS-P*J*) was transcriptionally induced under oxidative stress conditions and almost completely eliminated when the AlgR boxes 1 and 2 were mutated (pETS-P*J*-Δ*box1+2*) (Fig. 4A). This may indicate that during biofilm infection, AlgR is the primary transcription factor that regulates *nrdJ* expression under oxidative stress conditions. In addition, because biofilms show oxygen gradient heterogenicity, we wanted to study their pleiotropic regulation with the Anr transcriptional regulator. Anr is a transcription factor regulating several pathways related to oxygen tension and NO (20, 21). We previously mutated the Anr box in *nrdJ* (14) and, because it is an important regulator of oxygen-depletion conditions, we wanted to determine its involvement in *nrdJ* expression in biofilms under oxidative stress conditions. *nrdJ* expression was similar when the single AlgR boxes were mutated and when a double mutation was present in the AlgR and Anr binding sites (pETS-P*J*-Δ*box1+2*-Δ*Anrbox*), thus demonstrating that Anr does not play an important role during oxidative stress conditions (Fig. 4A).

Figure 4B shows that when the *P. aeruginosa* PAO1 Δ*algR* strain was complemented with AlgR or AlgR-D54N under oxidative stress conditions, *nrdJ* expression increased to the wild-type level. Furthermore, *nrdJ* induction was higher than that in the wild-type strain when the nonphosphorylated AlgR protein was used (Fig. 4B). These results indicate that AlgR plays a key role in *nrdJ* regulation during biofilm infection under oxidative stress conditions. *nrdJ* expression was quantified using COMSTAT software and plotted on a graph as shown in Fig. 4C.

The same expression pattern was observed when the expression of *nrdA* (class Ia RNR) was examined during biofilm formation (Fig. 5). Figure 5A shows images of

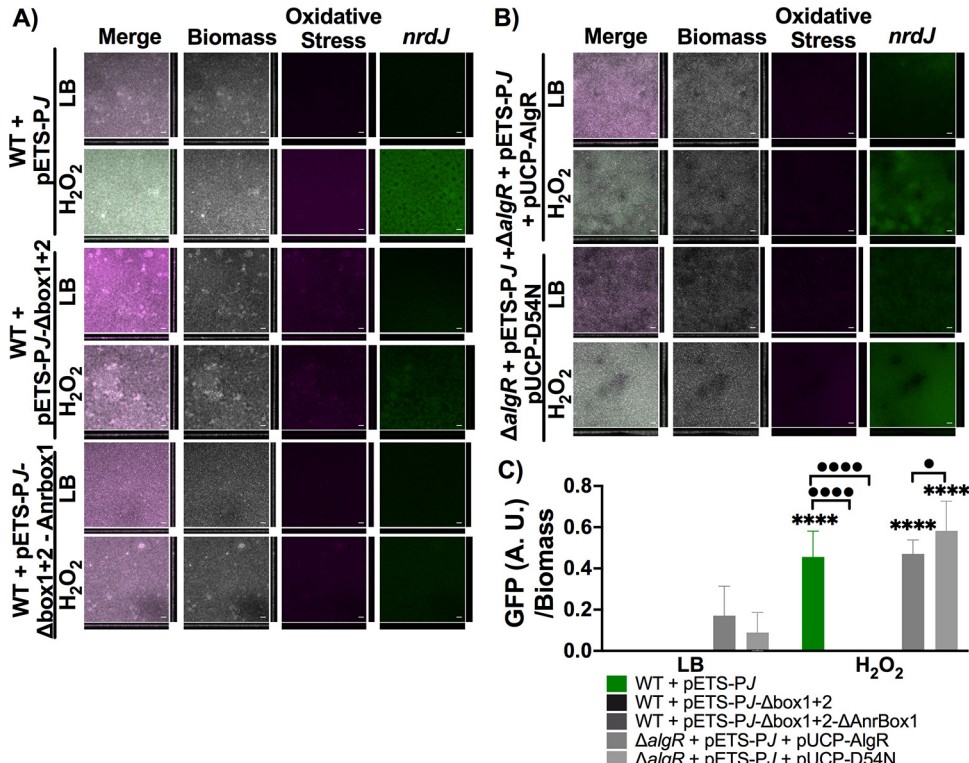

**FIG 4** *nrdJ* expression under oxidative stress conditions in a biofilm of *P. aeruginosa* PAO1. (A) Expression in the biofilm of *nrdJ* with the wild-type promoter, the AlgR boxes mutated, and the AlgR boxes and the Anr box mutated (P*J*, P*J*-Δ*box1+2*, and P*J*-Δ*box1+2*-Δ*Anrbox*, respectively). (B) Expression of *nrdJ* in a PAO1 Δ*algR* strain complemented with AlgR and AlgR-D54N. The samples were induced with 4 mM $H_2O_2$ or an equivalent volume of Luria-Bertani (LB) medium for 4 h. The biofilm was dyed with SYTO60 (gray) and CellROX Orange (magenta), and *gfp* is shown in green. Scale bars = 20 $\mu$M. (C) Data show the mean and standard deviation of *gfp* expression normalized by biomass. Student's unpaired *t* test was used to determine significant differences between the samples treated with $H_2O_2$ and their counterpart samples treated with LB medium (****, $P < 0.0001$) and significant differences between the strains induced with oxidative stress (●, $P < 0.05$; ●●●●, $P < 0.0001$).

*P. aeruginosa* PAO1 with pETS-P*A* and PAO1 with pETS-P*A*-Δ*box1* after the addition of LB medium supplemented with 4 mM $H_2O_2$ or pure LB medium. The images show that *nrdA* expression was induced under oxidative stress conditions during biofilm growth (Fig. 5C). The induction was removed when a promoter with a mutated AlgR box (pETS-PA-Δ*box1*) was used, demonstrating a direct role of AlgR in its transcriptional activation (Fig. 5A). When *P. aeruginosa* PAO1 Δ*algR* was complemented with AlgR or AlgR-D54N, it was observed that *nrdA* expression was restored after addition of $H_2O_2$ (Fig. 5B). The *nrdA* expression was not restored to the wild-type level but was closer when AlgR-D54N was used for complementation (Fig. 5C).

**nrdJ plays an important role during *Galleria mellonella* infection in response to oxidative stress.** *Galleria mellonella* is an outstanding alternative *in vivo* model to study bacterial infections because its innate response mimics that of mammals (22). *G. mellonella* has been previously used to study RNR expression during infection (23). Because *Galleria* larvae have autofluorescence, plasmids which produce GFP cannot be used to monitor gene expression during infection. Thus, the promoter regions used in this study were transcriptionally fused to *luxCDABE* genes that produce bioluminescence (23). After injecting the *Galleria* larvae, we made relative luminescence (RL) measurements for each larval group at 8, 14, 16, and 18 h postinfection. We determined the expression induction of each gene by comparing its expression at 14, 16, and 18 h with its expression at 8 h postinfection (the initial stage of infection).

Figure 6A shows the induction of *katA* and *katB* expression during infection. It was determined that the expression of *katA* (pLUX-P*katA*) was induced at 14, 16, and 18 h (113-, 865-, and 2,449-fold induction, respectively). In addition, *katB* (pLUX-P*katB*) was

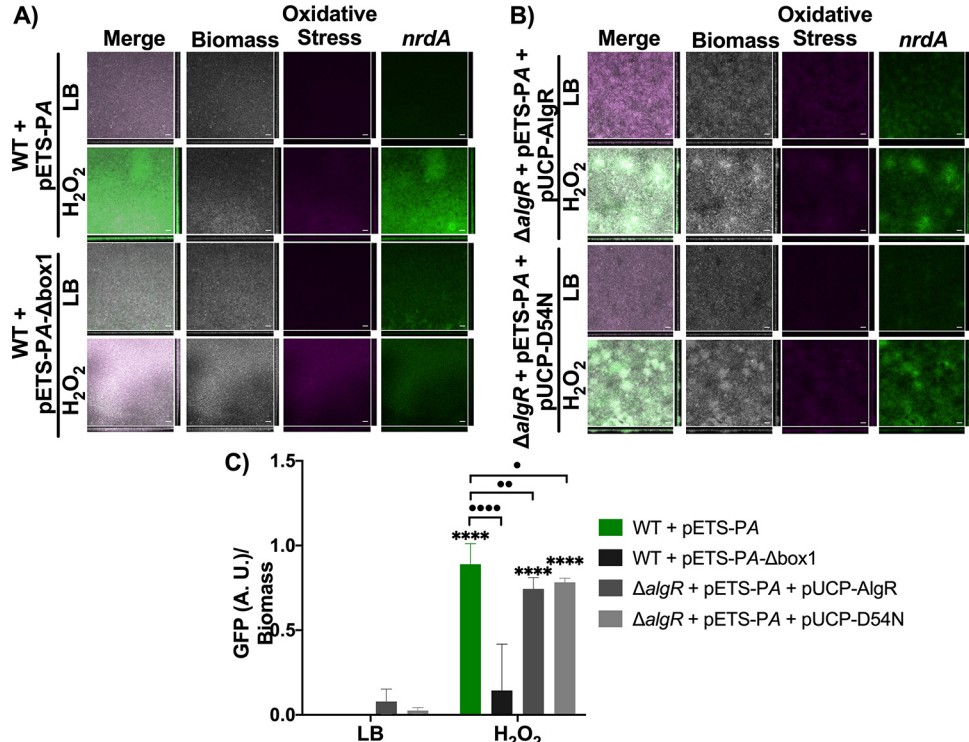

**FIG 5** *nrdA* expression under oxidative stress conditions in a biofilm. (A) Expression of the *nrdA* promoter and the *nrdA* promoter with the AlgR box mutated (P*A* and P*A*-Δ*box1*, respectively) in biofilm. (B) Expression of *nrdA* in a PAO1 Δ*algR* strain complemented with AlgR and AlgR-D54N. The samples were induced with 4 mM H$_2$O$_2$ or an equivalent volume of LB medium for 4 h. The biofilm was dyed with SYTO60 (gray) and CellROX Orange (magenta), and *gfp* is shown in green. Scale bars = 20 $\mu$M. (C) Data show the mean and standard deviation of *gfp* expression normalized by biomass. Student's unpaired *t* test was used to determine significant differences between the counterpart samples treated with H$_2$O$_2$ or LB medium (****, $P < 0.0001$) and significant differences between the strains induced with oxidative stress (•, $P < 0.05$; ••, $P < 0.01$; ••••, $P < 0.0001$).

induced at 14, 16, and 18 h, but its expression was lower than that of *katA* (51-, 85-, and 1,220-fold induction, respectively). These results may indicate that as the amount of ROS increases during infection, the expression of bacterial catalases increases as well to protect the bacteria from damage.

The induction of *algR* from the two promoters (pLUX-P*fimS* and pLUX-P*algR*) was measured (Fig. 6B). We found that the expression of P*algR* was higher than that of P*fimS* throughout the whole infection course, with the highest difference at 18 h postinfection (3,579- and 6,098-fold induction for P*fimS* and P*algR*).

We measured the expression of *nrdJ* (pLUX-P*J*), *nrdJ* with the AlgR boxes 1 and 2 mutated (pLUX-P*J*-Δ*box1+2*), *nrdJ* with the Anr box mutated (pLUX-P*J*-ΔAnrbox), and *nrdJ* with AlgR boxes 1 and 2 mutated and the Anr box mutated (pLUX-P*J*-Δ*box1+2*-ΔAnrbox) (Fig. 6C). The results showed that *nrdJ* expression was highly induced during infection, with *nrdJ* showing the highest induction at 18 h postinfection (130,257-fold induction). The expression of *nrdJ* decreased significantly when the AlgR boxes were mutated and when the Anr box was mutated (3,519- and 2,669-fold induction at 18 h, respectively). However, the most dramatic decrease in *nrdJ* expression was observed when both the AlgR and Anr boxes were mutated (825-fold induction). The expression observed when the AlgR boxes and the AlgR and Anr boxes were mutated was 158 times lower than that of the wild-type promoter, revealing the importance of both transcription factors during infection.

Finally, we measured the expression of the class I RNR gene *nrdA* with the WT promoter (pLUX-P*A*) and the promoter with the AlgR box mutated (pLUX-P*A*-Δ*box1*). The graph in Fig. 6D shows that the highest induction in *nrdA* expression occurred at 18 h

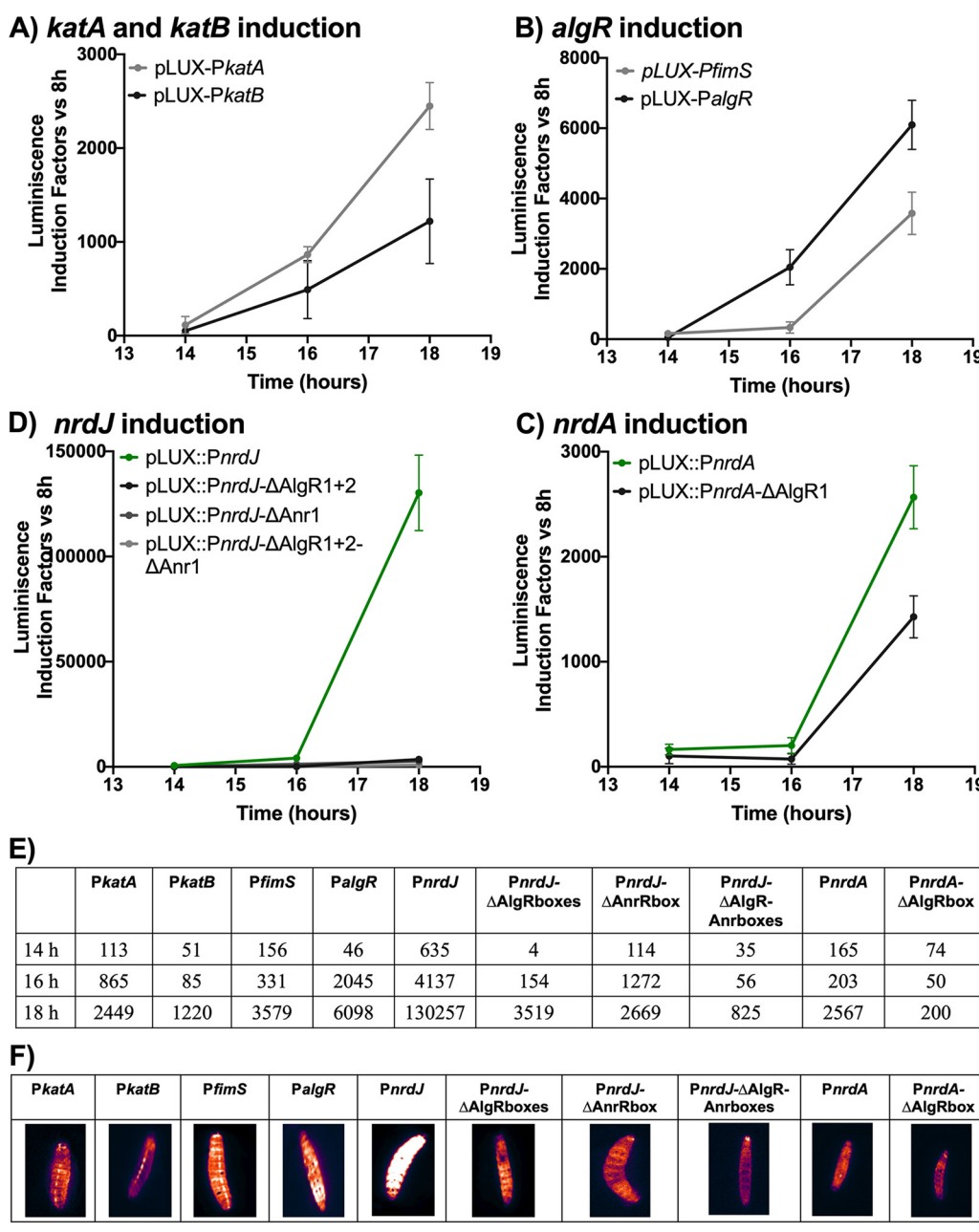

**FIG 6** Gene expression in *Galleria mellonella* larvae infection. Bioluminescence induction factors of expression in (A) *kat*, (B) *algR*, (C) *nrdJ*, and (D) *nrdA* promoters at different time points (14, 16 and 18 h postinfection). (E) Table listing bioluminescence induction factors. (F) Images of *G. mellonella* larvae bioluminescence taken with a ImageQuant LAS 4000 mini-imager.

postinfection (2,567-fold induction). At that time, the expression of *nrdA* with the AlgR box mutated was decreased by 13 times (200-fold induction).

Notably, the induction of *nrdJ* expression was much larger than that of *nrdA* expression (37 and 13 times, respectively, compared with that of the AlgR box mutant promoter counterpart), which demonstrates the key role of the class II RNR gene *nrdJ*, compared to *nrdA*, which was not the primary RNR gene activated during the infection of *G. mellonella*.

The bioluminescence produced by the different promoters was visualized using the ImageQuant LAS 4000, and images were taken at 18 h postinfection (Fig. 6E). The amount of bioluminescence shown in the images of the larvae depends on the expression of each gene, which is in accordance with the results shown in Fig. 6A to D.

## DISCUSSION

RNRs are essential enzymes in the life of any cell. Bacterial genomes commonly encode several RNRs to facilitate adaptation to different environmental conditions; thus, the expression and activation of each RNR class are tightly regulated. One of the main RNR transcriptional regulators is AlgR, which is part of the two-component system FimS-AlgR. AlgR is a global regulator, and FimS catalyzes its phosphorylation (8). The phosphorylation state of AlgR determines which regulatory pathways are activated or inactivated (3). We previously demonstrated that while *nrdA* is activated by phosphorylated AlgR in planktonic growth, *nrdJ* needs nonphosphorylated AlgR to be activated during early biofilm formation (8, 12).

The transcription of *algR* is carried out through two different promoters regulated by specific transcription factors (7). Environmental signals activate *algR* transcription through a specific promoter. In this study, we found that the expression of *algR* was induced under oxidative stress conditions, especially during the stationary growth phase (Fig. 1). The *algR* expression levels were not very high, probably due to its role as a global bacterial regulator. Global regulators are tightly regulated because dramatic changes in their expression can modify several metabolic pathways in the cell. However, although the activation was low, it was mainly observed through P*algR*, whose transcription is carried out after the binding of sigma factors such as RpoS and AlgU (7). The activation through the P*algR* promoter may indicate that AlgR does not depend on FimS when ROS are produced; thus, it may be nonphosphorylated or simply found in small amounts. Thus, new experiments using *rpoS* and *algU* mutant strains should be performed in the future to unravel their exact roles under oxidative stress conditions. Other studies have linked AlgR to the oxidative stress defense system in *P. aeruginosa*, but its role is not yet clear (19). It is possible that one of the main roles that AlgR plays against ROS is activating the *algD* operon to produce alginate, as it is known that alginate scavenges ROS, protecting bacteria (15). Due to this, studying *algD* or *algC* when ROS are present could be another acceptable way to delve into the roles of AlgR and alginate under oxidative stress conditions.

AlgR is a key factor in the regulation of class II (*nrdJ*) and class I (*nrdA*) RNRs and is involved in the control of the total dNTP pool in the cell. We determined that AlgR is responsible for inducing *nrdJ* expression under oxidative stress conditions (Fig. 2). It seems that *nrdJ* depends directly on AlgR binding, as its induction was removed when the AlgR binding boxes of the *nrdJ* promoter region were mutated and in a Δ*algR* strain (8). *nrdJ* induction was restored in a higher expression pattern when the *algR* mutation was complemented with the protein AlgR in its unphosphorylated state (pUCP-D54N). Using a *fimS* mutant strain could have been another acceptable way to study the phosphorylation state of AlgR under oxidative stress conditions. In the laboratory *P. aeruginosa* PA14 strain and the clinical isolate *P. aeruginosa* PAET1, we observed similar patterns of expression. *nrdJ* was induced when $H_2O_2$ was present, and this induction was abolished in a Δ*algR* strain and when using P*nrdJ*-Δ*box1+2*. AlgR-D54N restored *nrdJ* induction, indicating that AlgR was not phosphorylated. We observed that the basal expression of *nrdJ* in *P. aeruginosa* PA14 and PAET1 was lower than that in the PAO1 laboratory strain. We hypothesize that the reduced expression values observed may be due to differences in the genetic contexts of the three strains. However, the experiments confirmed that nonphosphorylated AlgR was the factor responsible for inducing *nrdJ* under oxidizing conditions.

When we measured the expression of *nrdA* under oxidative stress conditions, we found that class I RNRs were transcriptionally induced in *P. aeruginosa* PAO1, PA14, and PAET1 (Fig. 3) (8). This induction was absent in a Δ*algR* strain and when the AlgR binding box in the *nrdA* promoter was mutated (P*nrdA*-Δ*box1*). As shown in Fig. 2, *nrdA* expression in *P. aeruginosa* PA14 and in the clinical isolate *P. aeruginosa* PAET1 was lower than that in the laboratory strain PAO1. However, the experiments were useful enough to confirm that AlgR in its nonphosphorylated state was the factor responsible for inducing *nrdA* expression when $H_2O_2$ was present.

During infection, the production of ROS is one of the main defensive mechanisms against bacteria (15). To evaluate whether AlgR regulates RNR expression under oxidative

stress conditions during infection, we measured RNR induction in a continuous biofilm, which can simulate an infection-like situation (24). Other studies have already measured ROS in biofilms and have shown the importance of this growth condition during infection (25). In our experiments, we used a continuous flow biofilm to measure *nrdJ* and *nrdA* expression under oxidizing conditions (Fig. 4). The results showed that *nrdJ* expression was transcriptionally induced when $H_2O_2$ was present and that this induction was completely abolished when the AlgR boxes on the promoter region were mutated. When the Anr box found in the *nrdJ* promoter was mutated together with the AlgR boxes, we showed that *nrdJ* expression was even lower. These results demonstrate the critical role of AlgR in regulating *nrdJ* during biofilm formation. Anr is a transcription factor which regulates genes involved in anaerobic conditions (21), and it was shown to have a role in the regulation of *nrdJ* during biofilm formation (Fig. 4). However, the use of an *anr* mutant strain or a *nrdJ* promoter with the Anr box mutated could contribute to a better understanding of the role of Anr on its own. Because the complementation of Δ*algR* with AlgR and AlgR-D54N restored *nrdJ* expression under oxidative stress conditions, we determined that AlgR in its nonphosphorylated state was responsible for inducing *nrdJ* during biofilm formation. These results agree with those of previous studies showing that while phosphorylated AlgR regulates the initial steps of infection, nonphosphorylated AlgR controls the later steps, such as the production of alginate (7, 8).

In addition, we observed that the expression of *nrdA* was transcriptionally induced under oxidative stress conditions and that this induction was abolished when the AlgR box was mutated, confirming that AlgR is responsible for *nrdA* induction in biofilms. When we evaluated whether AlgR bound to the promoter in its phosphorylated or nonphosphorylated state, we observed that *nrdA* expression did not reach WT levels; however, AlgR-D54N produced slightly higher expression than wild-type AlgR (Fig. 5).

Finally, we used the *in vivo* model *G. mellonella* to measure the expression of several genes (Fig. 6). Other authors have linked the hemocytes of *G. mellonella* with the production of ROS inside the larvae (26). To test whether the bacteria inside the larvae could sense the oxidative stress produced during infection, we measured the expression of the *P. aeruginosa* catalases *katA* and *katB*. We observed that during *G. mellonella* infection, the expression of the constitutive catalase *katA* was increased at all the time points measured, and the expression of *katB*, the catalase whose expression is activated only when large amounts of ROS are detected (15), was activated at 16 and 18 h postinfection. After confirming that *P. aeruginosa* sensed the oxidative stress produced, we measured the expression of the remaining genes. We observed that *algR* expression was induced throughout the infection process, with higher levels of induction factors observed with P*algR* than with P*fimS*. When measuring the expression of the RNR, we observed that *nrdJ* showed the highest induction at 18 h postinfection, but its expression was completely eliminated when the AlgR boxes and the Anr box were mutated, confirming the remarkable roles of these transcription factors during infection (8, 21). The high induction of *nrdJ* at the latest time point suggests the important role of class II RNR in infections. However, when investigating the expression of *nrdA*, we observed that it decreased when the AlgR box was mutated, but it was not completely eliminated; thus, other factors may regulate the expression of *nrdA*. These results highlight the essential role of *nrdJ* in *G. mellonella* infection as the principal transcriptionally active RNR gene.

It is important to mention that all the experiments were conducted using plasmid constructions, which may not reflect the endogenous expression levels inside the cell and could yield certain modifications of the final levels depending on the system studied. However, our goal was not to investigate the specific amount of RNR production but to understand all actors taking place in their transcriptional regulation.

We have designed a schematic diagram to summarize the steps followed by *P. aeruginosa* under oxidative stress conditions (Fig. 7). Under nonoxidative conditions and after specific signals are sensed, the transmembrane protein FimS is autophosphorylated and phosphorylates the regulatory protein AlgR (3). Phosphorylated AlgR induces the

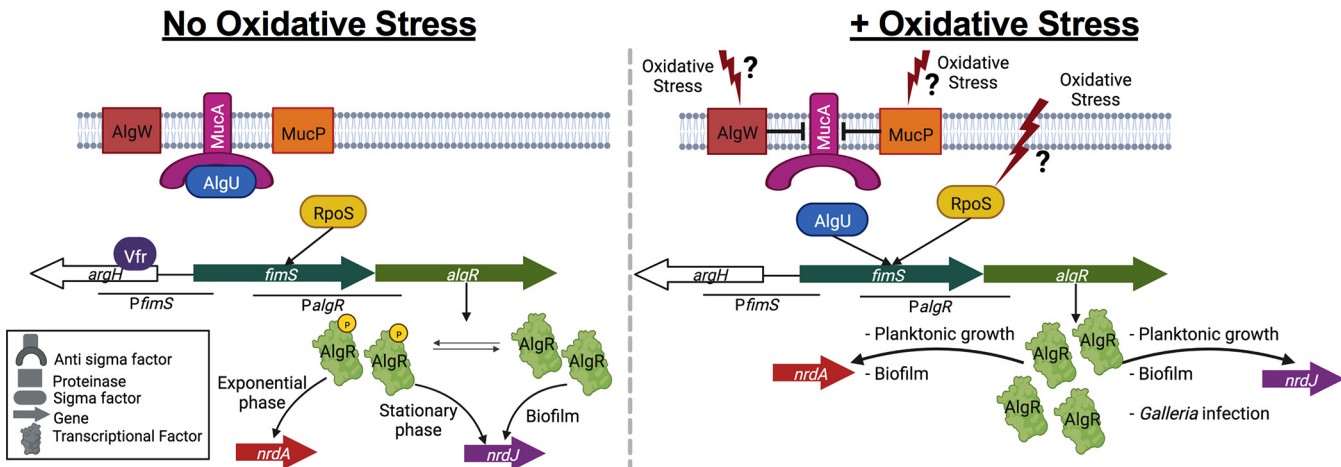

**FIG 7** Schematic representation of AlgR regulation with and without oxidative stress. Hypothetical regulatory model of the molecular pathway used by *P. aeruginosa* with or without oxidative stress conditions. Question marks indicate the putative pathways that oxidative stress may follow to induce *algR* expression and consequently class I (*nrdA*) and class II (*nrdJ*) ribonucleotide reductase (RNR) expression under specific growth conditions. Sources of information for each event are indicated in the manuscript. Artwork created with www.biorender.com.

expression of *nrdA* during the exponential growth phase and in the initial stages of biofilm colonization (8).

During the stationary growth phase, the sigma factors AlgU and RpoS bind to the promoter region of *algR*. AlgU is a sigma factor ($\sigma^{22}$) that is usually sequestered by the anti-sigma factor MucA. AlgU regulates genes important for alginate formation (27). RpoS is a sigma factor ($\sigma^{38}$) whose expression is cell cycle-dependent. It is regulated by cell density, and it regulates the transition to stationary phase (28). Once AlgU and RpoS bind to P*algR*, most of the AlgR protein produced is not phosphorylated (7). The nonphosphorylated AlgR favors the expression of *nrdJ* in the stationary phase of a planktonic culture and in biofilm.

Under oxidative stress conditions, AlgR expression is generated from P*algR*, and class I and class II RNRs are induced by nonphosphorylated AlgR in planktonic, biofilm and infection conditions (Fig. 1–6). These results suggest that AlgU and/or RpoS are the factors responsible for sensing ROS when they are present. AlgU has been linked with the oxidative stress defense system in *P. aeruginosa* (29). In addition, other proteins related to AlgU and MucA, such as AlgW and MucP, have been related to oxidative stress. We hypothesize that under oxidative conditions, AlgW or MucP cleaves MucA, freeing AlgU (15). The freed AlgU may activate *algR* expression, and AlgU and AlgR together may activate the transcription of their target genes. In addition, the RNR NrdA and NrdJ would produce the dNTPs needed to repair the damaged DNA. Once the ROS are eliminated by the multifaceted system of *P. aeruginosa*, AlgU would be sequestered by MucA again. On the other hand, the RpoS counterpart in *E. coli*, in addition to being involved in the transition to stationary phase, plays a role in the sensing of several stresses, including oxidative stress. However, no conclusive studies have confirmed a notable role of RpoS in oxidative stress-sensing in *P. aeruginosa* beyond a slight relationship (30). Taking all of this into account, we believe that AlgU is the sigma factor involved in the activation of *algR* under oxidative stress conditions. However, there are still many gaps that need to be filled. Experiments using Δ*algU* and Δ*rpoS* strains would help to shed light on the oxidative stress defense system and its relationship with the RNR regulatory network.

**Conclusion.** In conclusion, these results indicate that *algR* expression is induced after the addition of $H_2O_2$ in exponential and stationary phases and that *algR* expression under oxidative stress conditions is mostly obtained through the promoter P*algR*. These results may help unravel the role of AlgR in the oxidative stress response system. Additionally, we observed that under planktonic and biofilm conditions with oxidative stress, the expression of *nrdJ* and *nrdA* was transcriptionally induced by AlgR in its nonphosphorylated state in all *P. aeruginosa* strains tested. Finally, we showed that during

*G. mellonella* infection, the ROS produced by the larvae can be sensed by the bacteria and that *nrdJ* plays a major role in the production of dNTPs during an infection.

## MATERIALS AND METHODS

**Bacterial strains, plasmids, and growth conditions.** The bacterial strains and plasmids used are listed in Table S1. *E. coli* and *P. aeruginosa* strains were routinely grown in Luria-Bertani (Scharlab, Spain) medium at 37°C. Liquid cultures were shaken at 200 rpm. When necessary, antibiotics were added at the following concentrations: 50 $\mu$g/mL ampicillin and 10 $\mu$g/mL gentamicin for *E. coli*; and 100 $\mu$g/mL gentamicin, 40 $\mu$g/mL tetracycline, and 300 $\mu$g/mL carbenicillin for *P. aeruginosa*.

**DNA manipulation and plasmid construction.** Recombinant DNA manipulations were performed using standard protocols (31). The molecular biology kits and enzymes used in this study were purchased from Thermo Fisher Scientific (Spain) except as otherwise stated and were used following the manufacturers' instructions. DNA fragments were amplified using Phusion High-Fidelity DNA polymerase or DreamTaq Green PCR MasterMix with the primers listed in Table S2. DNA fragments were isolated from agarose gels using a GeneJet Gel Extraction kit. Plasmid DNA was extracted using a GeneJET Plasmid Miniprep kit and transferred into *P. aeruginosa* cells via electroporation using a Gene Pulser XCell electroporator (Bio-Rad) as previously described (32). All the constructs obtained were verified with DNA sequencing by Eurofins Genomics.

The plasmids pETS-P*katA* (pETS229), pETS-P*katB* (pETS230), and pETS-P*algR*-2 (pETS231) were constructed. Briefly, the *katA* (PA4236, 748 bp), *katB* (PA4613, 492 bp), and *algR* (PA5261, 878 bp) promoter regions were first amplified from *P. aeruginosa* PAO1 genomic DNA using primers 1 to 6, listed in Table S2. Each fragment was gel-purified, cloned into the pJET1.2b vector, and transformed into the *E. coli* DH5$\alpha$ strain. The resulting plasmids and the pETS130-GFP plasmid were digested with the corresponding restriction enzymes (BamHI-SmaI), and ligation was performed using the enzyme T4 ligase. These plasmids were electroporated into *P. aeruginosa* PAO1.

The Anr binding box in the *nrdJ* promoter region was mutated using PCR-based site-directed mutagenesis. The primer pairs 7/8 and 9/10 were used to amplify two fragments of the templates pETS-P*J* (pETS180) and pETS-P*J*-$\Delta$*box1+2* (pETS211) to generate the DNA fragments P*nrdJ*-$\Delta$*Anrbox* and P*nrdJ*-$\Delta$*AlgRbox1+2*-$\Delta$*Anrbox*, respectively. Each fragment was gel-purified and used as a template for a second round of PCR with the primers 7/10. The resulting amplicons were ligated into the pJET1.2b vector. The resulting plasmids and the pETS130-GFP plasmid were digested with BamHI-SmaI, and ligation was performed using the enzyme T4 ligase to obtain the plasmids pETS-P*J*-$\Delta$*Anrbox* (pETS232) and pETS-P*J*-$\Delta$*box1+2*-$\Delta$*Anrbox* (pETS233). These plasmids were electroporated into *P. aeruginosa* PAO1. Each construct was verified by DNA sequencing.

The promoter regions of the genes used in this study were cloned into the *lux* promoter vector pETS220. The plasmids pETS-P*A*-$\Delta$*box1* (pETS208), pETS-P*J*-$\Delta$*box1+2* (pETS211), pETS-P*J*-$\Delta$*Anrbox* (pETS232), pETS-P*J*-$\Delta$*box1+2*-$\Delta$*Anrbox* (pETS233), pETS-P*katA* (pETS229), pETS-P*algR*-1 (pETS207), and pETS-P*algR*-2 (pETS231) were digested using the enzymes SmaI and SacI. The fragments obtained were gel-purified and ligated using the enzyme T4 ligase into the digested SmaI-SacI pLUX (pET220) to obtain the plasmids pLUX-P*A*-$\Delta$*box1* (pETS234), pLUX-P*J*-$\Delta$*box1+2* (pETS235), pLUX-P*J*-$\Delta$*Anrbox* (pETS236), pLUX-P*J*-$\Delta$*box1+2*-$\Delta$*Anrbox* (pETS237), pLUX-P*katA* (pETS238), pLUX-P*algR*-1 (pETS240), and pLUX-P*algR*-2 (pETS241). The plasmid pETS-P*katB* (pETS230) was digested with the enzymes SmaI and SpeI, and the digested fragment was gel purified and ligated with the enzyme T4 ligase into the digested SmaI-SpeI pLUX to obtain pLUX-P*katB* (pETS239). These plasmids were electroporated into *P. aeruginosa* PAO1.

**Deletion of the *algR* gene in *P. aeruginosa* strain PA14 and the clinical PAET1 strains.** The plasmid pEX100Tlink was used to obtain *P. aeruginosa* PA14 and PAET1 *algR* mutant strains. First, we performed PCR to amplify the upstream and downstream regions of *algR* using the primer pairs 11/12 and 13/14 and the chromosomal DNA of PA14 and PAET1 as the templates. The amplified fragments were gel-purified and ligated into pJET1.2b using the enzyme T4 ligase. The plasmids obtained and the vector pEX100Tlink were digested using the restriction enzymes HindIII-BamHI and BamHI-SacI. The gel-purified fractions were ligated with the digested pEX100Tlink plasmid to obtain pEX100Tlink::*algR*'-'*algR* (pETS242) and pEX100Tlink::*algR*'-'*algR* (pETS243) for PA14 and PAET1, respectively. Afterward, the plasmids pETS242, pETS243, and pUCGmlox were digested with the restriction enzyme BamHI. The digested fractions were gel-purified and ligated with the enzyme T4 ligase to obtain the plasmids pEX100Tlink::*algR*'-Gmlox-'*algR* (PA14; pETS244) and pEX100Tlink::*algR*'-Gmlox-'*algR* (PAET1; pETS245). These final constructs were transformed into the *E. coli* S17.1 helper strain.

The PA14 $\Delta$*algR*Gm*lox* mutant (pETS131) and PAET1 $\Delta$*algR*Gm*lox* mutant (pETS133) of *P. aeruginosa* were generated by introducing pETS244 and pETS245, respectively, from *E. coli* S17.1 by conjugation. LB medium supplemented with 5% sucrose was used to counterselect the gentamicin-resistant transconjugants PA14 $\Delta$*algR*Gm*lox* and PAET1 $\Delta$*algR*Gm*lox*. Next, the plasmid pCM157 was electroporated into PA14 $\Delta$*algR*Gm*lox* and PAET1 $\Delta$*algR*Gm*lox*. The mutant strains were grown on LB broth supplemented with tetracycline to remove the gentamicin resistance cassette via the expression of the *cre* recombinase (33). The pCM157 was then removed from the mutant strains by three successive growth cycles in LB broth without tetracycline. The selected PA14 $\Delta$*algRlox* mutant (pETS132) and PAET1 $\Delta$*algRlox* mutant (pETS134) of *P. aeruginosa* were sensitive to gentamicin and tetracycline.

**Green fluorescent protein gene reporter assay.** *P. aeruginosa* bacterial cultures were grown on LB without antibiotics at 37°C and 200 rpm to $OD_{550}$ = 0.5 (exponential phase) and $OD_{550}$ > 2 (stationary phase). Upon reaching the desired $OD_{550}$, three independent 1-mL samples of each strain were collected. The samples were centrifuged for 10 min at 5,000 rpm, and the cell pellets were fixed with 1 mL of freshly prepared phosphate-buffered saline (PBS) solution containing 2% formaldehyde and stored in

the dark at 4°C for 10 min. The samples were centrifuged again, and the pellets were resuspended in 1 mL PBS. The fluorescence of the samples was measured in 96-well plates (Costar 96-Well Black Polystyrene plate, Corning) in an Infinite 200 Pro Fluorescence Microplate Reader (Tecan, Switzerland). Three measurements were performed for each independent sample.

**Biofilm in flow cell chambers, microscopy, and image analysis.** Different bacterial cultures of *P. aeruginosa* were adjusted to an $OD_{550}$ of 0.3 and inoculated into a three-channel flow cell (DTU Systems Biology, Technical University of Denmark). LB medium supplemented with 0.2% glucose was pumped through the flow cells at a constant flow rate of 42 $\mu$L/min and channel using an Ismatec ISM 943 pump (Ismatec, Wertheim, Germany). After 96 h of growth, the *P. aeruginosa* biofilms were treated with $H_2O_2$ at 4 mM for 4 h. The biofilm was stained with 10 $\mu$M SYTO60 reagent (Thermo Fisher Scientific, Spain) for 30 min at room temperature to visualize the total biofilm biomass and with 5 $\mu$M CellROX Orange reagent (Thermo Fisher Scientific) for 20 min at room temperature to detect oxidative stress.

A Zeiss LSM 800 confocal laser scanning microscope with excitation wavelengths of 488, 561, and 640 nm was used to generate the images. Microscope images were processed using ImageJ, and COMSTAT 2 software was used to quantify biofilm biomass as previously described (4, 34).

***Galleria mellonella* maintenance and injection.** *G. mellonella* larvae were fed a diet of 15% corn flour, 15% wheat flour, 15% infant cereal, 11% powdered milk, 6% brewer's yeast, 25% honey, and 13% glycerol and raised at 34°C in darkness as previously described (35). Before injection, the bacterial cultures of the different strains were centrifuged at 4,000 rpm for 10 min. The supernatant was discarded, and the pellet was washed three times in 5 mL PBS (Thermo Fisher Scientific, Spain). The culture was set to a final $OD_{590}$ of 1. Afterward, 10-fold serial dilutions of the equalized cultures were made with PBS. A total of 20 to 40 CFU of the bacteria were injected into groups of 6 *G. mellonella* larvae through the top right proleg using a 26-gauge microsyringe (Hamilton, NV, USA). The larvae were then kept at 37°C during the infection course.

**Bioluminescence measurements.** The relative luminescence of infected larvae was measured using a 6-well microtiter plate (Caplugs Evergreen, NY, USA) in an Infinite 200 Pro Fluorescence Microplate Reader (Tecan, Switzerland) with an integration time of 1,000 ms. GraphPad Prism 9.0 software (GraphPad Software, San Diego, CA, USA) was used to analyze the results obtained, and an ImageQuant LAS 4000 mini-imager (GE Healthcare, IL, USA) was used to obtain images of the chemiluminescence of the larvae at different time points postinfection with an exposure time of 30 s. The images were edited with ImageJ FIJI (version 1.52p, NIH, USA). Before each measurement, the larvae were anesthetized for 10 min on ice.

**Promoter alignment.** The T-COFFEE multiple sequence alignment server was used to align and compare the promoter regions of *nrdJ* and *nrdA* from the *P. aeruginosa* PAO1, PA14, and PAET1 strains (36).

**Statistical analysis.** GraphPad Prism 9.0 (GraphPad Software) was used to perform statistical analyses. Single comparisons were performed with unpaired or paired Student's *t* tests depending on the experiment type. The data values are expressed as the mean $\pm$ standard deviation.

## SUPPLEMENTAL MATERIAL

Supplemental material is available online only.

**FIG S1**, PDF file, 1.8 MB.
**TABLE S1**, PDF file, 0.1 MB.
**TABLE S2**, PDF file, 0.1 MB.

## ACKNOWLEDGMENTS

This study was partially supported by grants RTI2018-098573-B-I00 and PID2021-125801OB-100, funded by MCIN/AEI (10.13039/501100011033) and "ERDF: A way of making Europe," the CERCA program and AGAUR–Generalitat de Catalunya (2017SGR-1079), the European Regional Development Fund (FEDER), the Catalan Cystic Fibrosis Association, and Obra Social "La Caixa." A.R.-C. received a grant (PRE2018-083709) funded by MCIN/AEI (10.13039/501100011033) and "ESF: Investing in your future."

The manuscript was written by A.R.-C. E.T., L.P., and J.A. conducted the experiments. E.T. directed the research and revised the experimental data. All authors have approved the final version of the manuscript.

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
