## [Reviewer comments · mSystems]

Pseudomonas aeruginosa nonphosphorylated AlgR induces ribonucleotide reductase expression under oxidative stress infectious conditions

Alba Rubio-Canalejas, Joana Admella, Lucas Pedraz, and Eduard Torrents

Corresponding Author(s): Eduard Torrents, Institut de Bioenginyeria de Catalunya

Review Timeline:

Submission Date:	October 17, 2022
Editorial Decision:	December 8, 2022
Revision Received:	December 22, 2022
Accepted:	January 24, 2023

Editor: Laura Hug

Reviewer(s): The reviewers have opted to remain anonymous.

Transaction Report:

DOI: <https://doi.org/10.1128/msystems.01005-22>

December 8, 2022

Dr. Eduard Torrents
Institute for Bioengineering of Catalonia
Bacterial Infection and antimicrobial therapies
Baldiri Reixac 15-21
Barcelona, Barcelona 08028
Spain

Re: mSystems01005-22 (Pseudomonas aeruginosa nonphosphorylated AlgR induces ribonucleotide reductase expression under oxidative stress infectious conditions)

Dear Dr. Eduard Torrents:

Thank you for submitting your manuscript to mSystems. We have completed our review and I am pleased to inform you that, in principle, we expect to accept it for publication in mSystems. However, acceptance will not be final until you have adequately addressed the reviewer comments. Please note I am not requiring that you complete the additional experiments requested by Reviewer #1, though I agree they would strengthen the study. Certainly your revision must address the limitations that are inherent in your study given the lack of these experiments .

Preparing Revision Guidelines

Sincerely,

Laura Hug

Editor, mSystems

Reviewer comments:

Reviewer #1 (Comments for the Author):

Major comments:

Fig 1. While the statistical analysis the investigators use suggest a statistical significance of algR expression, the results are not impressive. As the algR promoter that appears to be sensitive oxidative stress is also controlled by AlgU/T and RpoS, testing of an algU/T and/or a rpoS mutant would help with their argument that algR expression is responsive to oxidative stress given that there is not even a 1.5-fold increase in expression. What if the entire algR promoter region was tested? Might this provide a more dramatic increase in expression given that both the algR and algZ/R promoters were increased in exponential phase? Additionally, their use of katA and katB reporters during exponential phase show the most dramatic increase in expression whereas in stationary phase the difference is not impressive.

While I agree that the conditions the investigators used during exponential conditions were capable of activating the ROS response, it is not convincingly demonstrated that algR expression is responding to oxidative stress. A possible alternative conclusion to the researchers' results is that it is AlgR activity that is activated by oxidative stress. Testing of a promoter, such as algC, might help to shed light on this.

Fig 2+3. While the D54N complementation resulted in the largest increase in both nrdJ and nrdA it is difficult to interpret these experiments without knowing that both the wild-type and the D54N algR alleles had similar expression levels. Testing the reporter fusions in an algZ mutant would help to provide support for the researchers claim. The inclusion of multiple strains lends support to the conclusion that the regulation of the nrdJ and nrdA promoters is similar among different strains.

Fig 4+ 5. The biofilm data shows the relevance of AlgR controlling nrdJ and nrdA during oxidative stress in a biofilm. For Figure 4 on line 501 the conclusion that ANR is involved in regulating nrdJ should be changed or a Δ anr strain tested with nrdJ expression or a promoter fusion containing a mutated ANR box. On line 36, I do not understand the relevance of the PilJ sentence.

Figure 6. I do not think the Galleria infection model can be used to make conclusions about chronic infections. (lines 527+528).

Figure 7. The model suggests that algD and nrdJ expression are activated under their system conditions. Has this been tested? Testing of an algD reporter would shed light on the model being tested and should be incorporated into this study if this is the model. Additionally, it might be relevant to test an Δ algU and/or Δ rpoS strain to confirm the working model. At least some invitro studies examining nrdJ and nrdA expression would be helpful.

Minor comments:

Line 50: reword sentence "no alginate production"

Line 73: I don't think this statement is accurate. "phosphorylated AlgR activates the T3SS" ? See Jones et al. 2010 J Bact. Activation of the Pseudomonas aeruginosa AlgU regulon through mucA mutation inhibits cAMP/Vfr signaling.

Line 112-3: "We discovered that ROS are produced during Galleria melonella infection" This was already known

Line 313: "was" is italicized and shouldn't be

Line 501: This conclusion should be modified or data showing only an anr mutant strain provided.

Line 536: I'm not sure of the relevance of the PilJ sentence. How does PilJ "send signals" by FimS?

Reviewer #2 (Comments for the Author):

The manuscript by Rubio-Canalejas seeks to establish the role of AlgR transcription factor in regulation of ribonucleotide reductase expression under oxidative stress. To do this, the authors generate a series of plasmid-based GFP reporters, with a range of promoters, to examine expression under a range of conditions.

The work is overall thorough and the results are consistent with the interpretations given. The main caveat is that this work is done exclusively with plasmid constructs (reporters plus promoters, and plasmid-based AlgR complementation). I'm a little surprised the authors don't mention this as a caveat, since higher expression levels from plasmids might not always reflect the endogenous levels and there is a chance that this could yield artefacts. I'd want to see that at least discussed. However, what has been presented represents a substantial amount of detailed work, and is worthy of publication with minor revision to address general limitations of this approach.

Reviewer #3 (Comments for the Author):

Rubio-Canalejas et al. investigate transcriptional regulation of class I and II ribonucleotide reductase (RNR) genes in different strains of *P. aeruginosa* in response to oxidative stress and during the infection of honeycomb moth *Galleria mellonella*. This is a very carefully executed study and the conclusions drawn by the authors are supported by the obtained results. One slight weakness of the paper is that the Discussion largely summarizes the data and does not put them in a bigger perspective. The authors state on p. 24, lines 554-555 "In addition, the RNR NrdA, but mostly NrdJ, increases the pool of dNTPs needed to repair the damaged DNA", but no reference to support this statement is given. Perhaps it would be worthwhile to discuss that DNA damage activates RNR in other bacteria and unicellular organisms, leading to increases in dNTP concentrations, and that elevated dNTP pools help DNA polymerases to bypass DNA lesions. To my knowledge, there is no data in the literature suggesting that increases in dNTP pools are needed to repair the damaged DNA, but if the authors are aware of such studies, it would be important to provide references supporting this statement.

Minor points:

1. The authors sometimes write that promoters are "transcriptionally fused" to reporter genes, and sometimes that promoters are just "fused" to reporter genes. Is there a difference? Is it common to fuse promoters to reporter gene non-transcriptionally?
2. Line 285, move (8) to the end of the sentence?
3. No specific contribution is listed for Lucas Pedraz. Did this co-author only approve the final version of the manuscript?

December 21st, 2022

Dear Prof. Laura Hug,

Thank you for thoroughly reviewing our paper and for the opportunity to submit a revised version. We much appreciate the reviewer's constructive comments on our manuscript (Manuscript mSystems01005-22, *Pseudomonas aeruginosa nonphosphorylated AlgR induces ribonucleotide reductase expression under oxidative stress infectious conditions*), which have been of great help and have improved the manuscript over the previous version. We are really thankful for their work on our manuscript.

Our responses to his/her comments are detailed below (in red).

With the manuscript changes detailed below and our answers to the reviewer's comments, we hope you will now find the revised version of our manuscript acceptable for publication in **mSystems**.

Sincerely,

Dr. Eduard Torrents

Reviewer comments:

Reviewer #1 (Comments for the Author):

Major comments:

Fig 1. While the statistical analysis the investigators use suggests a statistical significance of algR expression, the results are not impressive. As the algR promoter that appears to be sensitive oxidative stress is also controlled by AlgU/T and RpoS, testing of an algU/T and/or a rpoS mutant would help with their argument that algR expression is responsive to oxidative stress given that there is not even a 1.5-fold increase in expression. What if the entire algR promoter region was tested? Might this provide a more dramatic increase in expression given that both the algR and algZ/R promoters were increased in exponential phase?

We thank the reviewer for this suggestion. We completely agree with the reviewer that the algR expression levels are not very high. However, as AlgR is a global metabolic regulator, its expression should be tightly regulated, avoiding high expression changes as significant AlgR production alters the transcription of several critical pathways in the cell with dramatic changes in the bacteria (see introduction section on page 4 and 5, and discussion section on page 20, lines 466-469).

We also agree that further studies are required to study the specific influence of algU or rpoS on algR production, and indeed their mutants may help to understand their role under oxidative stress conditions. However, we will continue work in this direction, and it is not right now the scope of this manuscript, as our main goal was to study the RNR expression concerning oxidative stress and the transcription factor AlgR. However, we entirely agree with the reviewer and have added several sentences highlighting this limitation in the paper's discussion section. Please, see the new modifications on page 20, lines 472-474, and page 24, lines 578-579.

Additionally, their use of katA and katB reporters during exponential phase show the most dramatic increase in expression whereas in stationary phase the difference is not impressive.

We appreciate the reviewer for pointing this out and agree with his or her observation. We used katA and katB only as controls during our experiments to test the oxidative stress conditions. The fact that their induction is lower in stationary phases may be due to many reasons, as Pseudomonas encodes several pathways to eliminate ROS. However, we believe that katA and katB serve their purpose in our study.

While I agree that the conditions the investigators used during exponential conditions were capable of activating the ROS response, it is not convincingly demonstrated that algR expression is responding to oxidative stress. A possible alternative conclusion to the researchers' results is that it is AlgR activity that is activated by oxidative stress. Testing of a promoter, such as algC, might help to shed light on this.

We thank the reviewer for this suggestion. It is already known that alginate protects bacteria from ROS, which certainly using an algC transcriptional fusion could be an excellent way to study the role of AlgR and alginate under oxidative stress conditions. However, as this work focuses on understanding transcriptional regulation or RNR genes under oxidative stress, we believe that studying the RNR promoters when the AlgR boxes are mutated under oxidative stress conditions is also an excellent way to see the AlgR role under oxidative stress conditions. Nevertheless, as suggested by the reviewer, we have included the possibility of studying algC promoter in the discussion section. Please, see the new modifications on page 20, lines 477-479.

Fig 2+3. While the D54N complementation resulted in the largest increase in both *nrdJ* and *nrdA* it is difficult to interpret these experiments without knowing that both the wild-type and the D54N *algR* alleles had similar expression levels. Testing the reporter fusions in an *algZ* mutant would help to provide support for the researchers claim. The inclusion of multiple strains lends support to the conclusion that the regulation of the *nrdJ* and *nrdA* promoters is similar among different strains.

*We appreciate the reviewer's observation, and we agree with his/her comment that using an *algZ* mutant could have been another way to study the phosphorylation state of AlgR. However, it is known that the D54N mutation avoids AlgR phosphorylation (see references 3, 7 and 8 in our manuscript), and we believe that it is a reasonable assumption that the expression levels of AlgR and AlgR-D54N were alike as the coding genes for both proteins were cloned in the same vector and used to complement the *algR* mutation. However, we have included a sentence in the text to clarify this point. Please, see the modifications on lines 486-487.*

Fig 4+ 5. The biofilm data shows the relevance of AlgR controlling *nrdJ* and *nrdA* during oxidative stress in a biofilm. For Figure 4 on line 501 the conclusion that ANR is involved in regulating *nrdJ* should be changed or a Δanr strain tested with *nrdJ* expression or a promoter fusion containing a mutated ANR box. On line 36, I do not understand the relevance of the PilJ sentence.

*We agree with the suggestions provided by the reviewer. We have removed the PilJ sentence as it was confusing, and we have added a comment explaining the need to study *Anr* expression using an *anr* mutant strain or a *nrdJ* promoter with the *Anr* box mutated. Please, see the new modifications on page 22, lines 512-516.*

Figure 6. I do not think the Galleria infection model can be used to make conclusions about chronic infections. (lines 527+528).

Thank you for pointing this out. We have changed the sentence. Please, see the modifications on lines 541-542.

Figure 7. The model suggests that *algD* and *nrdJ* expression are activated under their system conditions. Has this been tested? Testing of an *algD* reporter would shed light on the model being tested and should be incorporated into this study if this is the model. Additionally, it might be relevant to test an $\Delta algU$ and/or $\Delta rpoS$ strain to confirm the working model. At least some invitro studies examining *nrdJ* and *nrdA* expression would be helpful.

We thank the reviewer for pointing this out. We have modified the text to make clear that the model is just a hypothesis of what may be happening based on the results obtained in the study. We believe that in this way, the text will not lead to misunderstandings. We have also included the need to perform experiments using an $\Delta algU$ and/or $\Delta rpoS$ strains. Please, see modifications on page 24, lines 569-572 and 578-579.

Minor comments:

Line 50: reword sentence "no alginate production"

Line 73: I don't think this statement is accurate. "phosphorylated AlgR activates the T3SS"? See Jones et al. 2010 J Bact. Activation of the Pseudomonas aeruginosa AlgU regulon through mucA mutation inhibits cAMP/Vfr signaling.

Line 112-3: "We discovered that ROS are produced during Galleria melonella infection" This was already known

Line 313: "was" is italicized and shouldn't be

We thank the reviewer for these suggestions. We agree, and we have modified all these points accordingly. See modifications on lines 63, 87, 126, and 321.

Line 501: This conclusion should be modified or data showing only an anr mutant strain provided.
Line 536: I'm not sure of the relevance of the PilJ sentence. How does PilJ "send signals" by FimS?

We thank the reviewer for pointing this out. We have already answered these suggestions in Fig 4+5 significant points.

Reviewer #2 (Comments for the Author):

The manuscript by Rubio-Canalejas seeks to establish the role of AlgR transcription factor in regulation of ribonucleotide reductase expression under oxidative stress. To do this, the authors generate a series of plasmid-based GFP reporters, with a range of promoters, to examine expression under a range of conditions.

The work is overall thorough and the results are consistent with the interpretations given. The main caveat is that this work is done exclusively with plasmid constructs (reporters plus promoters, and plasmid-based AlgR complementation). I'm a little surprised the authors don't mention this as a caveat, since higher expression levels from plasmids might not always reflect the endogenous levels and there is a chance that this could yield artefacts. I'd want to see that at least discussed. However, what has been presented represents a substantial amount of detailed work, and is worthy of publication with minor revision to address general limitations of this approach.

We appreciate the reviewer's suggestion and agree with his/her observation. Thus, we have modified the text to exhibit the limits attached to using plasmids in our study. Please, see modifications on page 23, lines 546-550.

Reviewer #3 (Comments for the Author):

Rubio-Canalejas et al. investigate transcriptional regulation of class I and II ribonucleotide reductase (RNR) genes in different strains of *P. aeruginosa* in response to oxidative stress and during the infection of honeycomb moth *Galleria mellonella*. This is a very carefully executed study and the conclusions drawn by the authors are supported by the obtained results. One slight weakness of the paper is that the Discussion largely summarizes the data and does not put them in a bigger perspective. The authors state on p. 24, lines 554-555 "In addition, the RNR NrdA, but mostly NrdJ, increases the pool of dNTPs needed to repair the damaged DNA", but no reference to support this statement is given. Perhaps it would be worthwhile to discuss that DNA damage activates RNR in other bacteria and unicellular organisms, leading to increases in dNTP concentrations, and that elevated dNTP pools help DNA polymerases to bypass DNA lesions. To my knowledge, there is no data in the literature suggesting that increases in dNTP pools are needed to repair the damaged DNA, but if the authors are aware of such studies, it would be important to provide references supporting this statement.

We thank the reviewer for pointing this out. It is a hypothesis based on the results obtained in this study. We have modified the text to clarify, so it does not lead to misunderstandings. When new DNA is synthesized (de novo or because it is damaged), there need to be dNTPs for this synthesis. The RNR are the enzymes responsible for dNTPs production. We have also modified the discussion to include some of the limitations of our study. Please, see the modifications on pages 24, lines 570-571.

Minor points:

1. The authors sometimes write that promoters are "transcriptionally fused" to reporter genes, and sometimes that promoters are just "fused" to reporter genes. Is there a difference? Is it common to fuse promoters to reporter gene non-transcriptionally?

We appreciate the reviewer's observation. All the promoters were transcriptionally fused to the GFP. We have modified the text to include the word "transcriptionally" every time. Please, see the modifications on lines 298 and 301.

2. Line 285, move (8) to the end of the sentence?

Thank you for pointing this out. Please, see the modification on line 294.

3. No specific contribution is listed for Lucas Pedraz. Did this co-author only approve the final version of the manuscript?

Thank you for this comment. Yes, Dr. Lucas Pedraz approved the final version of the manuscript.

January 24, 2023

Dr. Eduard Torrents
Institut de Bioenginyeria de Catalunya
Bacterial Infection and antimicrobial therapies
Baldiri Reixac 15-21
Barcelona, Barcelona 08028
Spain

Re: mSystems01005-22R1 (Pseudomonas aeruginosa nonphosphorylated AlgR induces ribonucleotide reductase expression under oxidative stress infectious conditions)

Dear Dr. Eduard Torrents:

I am pleased to inform you that your manuscript has been accepted, and I am forwarding it to the ASM Journals Department for publication. For your reference, ASM Journals' address is given below. Before it can be scheduled for publication, your manuscript will be checked by the mSystems production staff to make sure that all elements meet the technical requirements for publication. They will contact you if anything needs to be revised before copyediting and production can begin. Otherwise, you will be notified when your proofs are ready to be viewed.

If you would like to submit a potential Featured Image, please email a file and a short legend to msystems@asmusa.org. Please note that we can only consider images that (i) the authors created or own and (ii) have not been previously published. By submitting, you agree that the image can be used under the same terms as the published article. File requirements: square dimensions (4" x 4"), 300 dpi resolution, RGB colorspace, TIF file format.

We recognize that the video files can become quite large, and so to avoid quality loss ASM suggests sending the video file via <https://www.wetransfer.com/>. When you have a final version of the video and the still ready to share, please send it to mSystems staff at msystems@asmusa.org.

Sincerely,

Laura Hug
Editor, mSystems

Journals Department
E-mail: mSystems@asmusa.org